# Case Study: Teaching with Industry (TWI) Using New Videoconferencing Technology and Innovative Classroom Setups

Francois Jacobs *, William Cain, Renxiang Lu and Amy Daugherty

Department of Civil and Architecture Engineering and Construction Management, University of Wyoming, Laramie, WY 82071, USA; wcain@uwyo.edu (W.C.); rlu@uwyo.edu (R.L.); adaughe2@uwyo.edu (A.D.)
* Correspondence: fjacobs@uwyo.edu; Tel.: +1-307-766-2390

**Abstract:** This paper describes a case study of a novel teaching method where the "Teaching with Industry" (TWI) model–industry practitioners incorporated as co-instructors in a semester-long classroom setting–is enhanced by using new videoconferencing technologies such as Zoom and Meeting Owl Pro, and innovative classroom setups. This enhanced model was developed with the intent to bridge the gap between information acquired in the classroom and the skills and competencies required in the industry. The different teaching platforms not only facilitated the teaching when industry practitioners were/are not able to be physically present in the classroom, but also led to efficient organization of the different activities carried out in class. Results obtained from end-course surveys showed that students had a positive experience using *Zoom* and *Meeting Owl Pro* welcoming the opportunity to engage with industry practitioners and gain better understanding of the practical usefulness of the course.

**Keywords:** industry practitioners; co-instruction; integrated videoconferencing technology; shared teaching platform

## 1. Introduction

Education-to-work transitions are a difficult time for students to navigate, often in part because of the differences in expectations between the two domains [1]. The main factor causing this disconnect is that academic faculties and industry practitioners differ in their perceptions of the characteristics of a learning environment that leads students to be successful in their future careers [2]. In academia, since instructors of record are normally researchers, the teaching philosophy is based on theoretical concepts that can be investigated for further contribution. However, employers in the industry expect their workers to be able to practice what they learned in real-time with the objective of making profit for the company [3].

The gap between classroom instruction and industry requirements can be especially noticeable in career and technical education (CTE) degrees, like Construction Management. Much of what is expected and required in students of CTE degree programs may not translate easily into desired skill sets for employers. Industry recruiters observed that although new hires generally have sufficient technical skills, many do not have a good grasp of the actual applicability of these skills in an actual project [4]. Others seem to have lack of soft skills such as critical-thinking, leadership, and communication, which are as important as the technical competencies [5]. Some employers believe that the practices taught in academia are obsolete and too didactic which have little practical use in the practice of the profession [6]. Thus, a small group of employers are even willing to hire experienced personnel without a four-year college degree in lieu of freshly graduated students without experience [7]. As a matter of fact, many students do not feel prepared to transition to the professional market. A great number think that the

theoretical component learned in higher education is not enough to gain advantage in the construction industry [8]. Also, students with industry work experience reported that they did not have the capabilities at the beginning to carry out some of the proposed tasks on the jobsite [9]. Many needed additional training before being assigned to actual tasks from their representing companies [10]. As a result of this dichotomy, it is essential for university programs to incorporate pedagogical strategies to aid students in developing desirable and necessary skills more effectively. One of these strategies is to balance a combination of meaningful theoretical content in the classroom with guidance and insight from industry practitioners.

The objective of this paper is to detail a case study class in which the TWI model was adopted and supported by using new videoconferencing technologies (*Zoom* and *Meeting Owl Pro*) and innovative classroom setups. This study highlights the different elements for the design and implementation of the TWI model including the entities involved, technologies, and direct (quizzes and reflection papers) and indirect assessment (end-course survey) results obtained from the students and industry practitioners.

## 2. Literature Review

The construction sector has always had a great influence on the United States Gross Domestic Product (GDP) and employment. In 2019, 6.4% of all industry employment and 4.1% of the GDP were attributable to this sector [11]. Due to the continuous increase of the population in residential areas, the demand for new buildings, roads, and other infrastructures, propelled this sector to grow even further. With these new constructions put in place, it is forecast that the expenditure in the construction sector will reach over $1.53 trillion by 2022 [12], which is significantly higher than the contribution over $1 trillion reported in 2008 [13]. As a result of this increase in demand, the overall employment in the construction sector is projected to grow 6% from 2020 to 2030, and the employment of construction managers is projected to double between the same period [14]. Besides the changes in size, the requirements of the construction industry have also evolved in complexities as construction laws, workplace safety, and environmental issues are becoming the new challenges in this sector [15]. Thus, it is vital for construction programs to provide an education that prepares students to this new reality.

To minimize the divergence between academia and industry described above, instead of completely redesigning the academic curricula, an efficient approach to solve this problem is to have industry practitioners collaborating in activities organized in higher education. Existing literature shows that there have already been different initiatives in construction education, in which industry participation was integrated. Some of these include providing students with financial support [16], arranging organized field trips [17], advertising internship opportunities [18], sponsoring student competitions [19], organizing collaborative conferences [20], developing joint research projects [21], and other creative initiatives.

Besides the collaborations aforementioned, the initiative that has the most impact on the students' connection to industry is the integration of industry practitioners in the academic teaching. It is quite prevalent nowadays to invite industry practitioners to give presentations about industry life to students [22]. Also, programs sometimes hire industry practitioners during summer to teach short-courses or host boot camps where topics pertaining to the practice of construction are approached [23]. To further integrate industry participation in higher education, some programs incorporate industry practitioners to co-teach courses with the instructor of record (TWI). In the TWI model, the original curriculum and assessment methods are not significantly altered with the inclusion of industry practitioners as co-instructors. Rather, the intent of this model is to have the original curriculum interpreted by the industry practitioners in ways they feel would be similar to the industry perspectives which they represent and give voice [24]. Although deliverables and material content can be requested from the instructor of record, it is the entire responsibility of the industry practitioners to conduct the class as they see fit [25].

Despite the lack of experience in educational teaching, students generally enjoy classes taught by industry practitioners since many real-life examples and insights about the practice of professions are discussed [26]. By having a close interaction with the students, the industry also benefits from this initiative by having priority in recruiting the most preeminent students to their companies [27].

The integration of industry practitioners in academia can be supported by videoconferencing technologies especially now, as in-person meetings are restricted during the COVID-19 pandemic. Also, the use of videoconferencing allows the industry practitioners to conduct classes remotely, as they are still responsible to their employers for their daily on-site work tasks [28]. Although this may have been viewed as an inconvenience in the past, remote communication systems are now one of the most reliable technologies in the world [29]. Due to the globalization and consequent necessity of national and international networking, every school and company are equipped with the most cutting-edge devices (e.g., computers, I-Pads, smartphones, etc.) and connected through the most up-to-date internet systems, which make remote interaction almost as if in-person. On top of that, all these devices are compatible with software such as *Zoom*, *Skype* and, *TEAMS* which not only enhance the audio and visual quality but also have user-friendly platforms [30]. Thus, remote classrooms have become commonplace.

The results of training and education delivered by remote instruction have shown that students perform better using an online platform. One explanation for this result is that students become more autonomous learning the delivered content and in a more continuous fashion when using the online platform, while in face-to-face classes, students believe that what is learned in class is already enough [31]. In another example, it has been observed that the passing grade of a higher education course was higher when it is delivered online than in a face-to-face classroom. It could be that the students' interest with technology further motivates them to do well in the class [30]. Additionally, students in remote classrooms seem to benefit in many other aspects: stimulation of critical thinking during group project discussions [32], gains in confidence during presentations [33], increase of motivation in class participation [34], receipt of immediate feedback from instructors [35], and be less likely distracted by colleagues [36].

Another element that further improves the TWI model is the adoption of the right classroom physical environment. Past research demonstrated that classrooms' structural features such as lightning, acoustics, and temperature, and symbolic features like décor and signs, can positively or negatively affect students' integration and performance in class [37]. It was observed that students tend to perform worse (lower test score) at institutions with structural inadequacies such as damaged plumbing, broken windows, and ventilation problems [38]. Also, subtle messages shown on paintings and boards exhibited in schools may sometimes have a negative impact especially on students of color and female students [39]. As a result, policymakers in many schools in the United States are putting in effort to address these problems that may prevent students from feeling integrated and achieving academic success.

In particular, a subcategory of the classroom physical environment that plays a big role on the students' learning experience is the classroom setup. It has been noted that the arrangement of chairs and tables influences the students' participation and sense of control in a classroom [40]. Although it may depend on the characteristics of each student (e.g., personality, gender, age, etc.) studies revealed that different classroom setups may be suitable for different learning goals [41]. The traditional classroom layout is to have tables displayed in rows, with a maximum of two students per table facing the instructor. This setup is efficient for students to concentrate on the lectures as well as to perform individual tasks [42]. However, with the implementation of new activities in education, the row layout was shifted to small groups either in a circle or semi-circle. The former group layout allows students to discuss course topics and learn from each other through social interaction [43]. Unlike the circle setup which stimulates more interaction within the group, the semi-circle setup allows the groups to communicate directly to the instructor [44].

Despite the advantages of the group setup, it is recommended to adopt it only for purposes of social interaction because this setup tends to lead to disruptive and off-task behaviors in the classroom [45].

Although there are many studies that detail the participation of industry practitioners in academia, use of new videoconference technology in teaching, and integration of different classroom setups, this case study is unique as it integrates the benefits of these three components together.

## 3. Methodology and Procedure

This unfunded case study looks at the remote components of the Construction Management program of the University of Wyoming. The design framework of the TWI model is supported by new videoconferencing technologies (*Zoom* and *Meeting Owl Pro*) and innovative classroom setups. The elements of the framework are composed of the following:

Element 1–Course and Participants
Element 2–Research Platform (videoconferencing technologies and classroom setups)
Element 3–Direct Assessment (evaluation of students' performance)
Element 4–Indirect Assessment (feedback from students and industry practitioners)

Each of the elements is detailed below. The elements are integrated in a continuous cycle so that the TWI model can be continuously improved in the future (Figure 1).

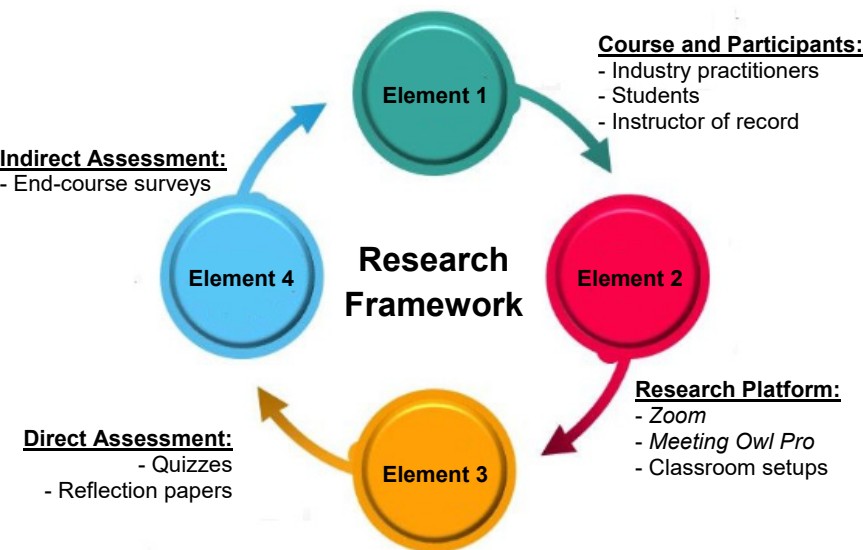

**Figure 1.** Research design framework.

### 3.1. Course and Participants

This case study took place in the Construction Management program hosted in the College of Engineering at the University of Wyoming. In a 16-week semester-long Construction Management course, the instructor of record invited industry practitioners to be part of the course in order to establish a connection between theoretical coursework and required industry skills. Since conducting a case study across the full spectrum of a four-year degree program was not practical, this study only evaluates the TWI model supported by new videoconferencing technologies and innovative classroom setups for the CM 2300: Construction Safety 3-credit course taught in the semester of Spring 2021.

The class is composed of a total of 60 students and was organized around fifteen chapters of a selected textbook: thirteen of which were taught in a traditional fashion (by the instructor of record), and two of which were co-taught by different industry practitioners who specialize in those topics. A total of six industry practitioners were involved in this TWI model. In this co-instructor agreement, the instructor of record was only responsible for the introduction of the chapter, and the industry practitioners were given the

freedom to develop full-course presentations (e.g., PowerPoint slides and classroom deliverables). The industry practitioners were also encouraged to provide additional content related to their own experiences so that students can get to know more about the construction industry. Since the industry practitioners still needed to exert their professional activities, they conducted their class by videoconferencing in real-time via *Zoom* while the students were in the classroom. As a reference for the students in class, the information regarding to the occupation and affiliation of all the industry practitioners were listed in the syllabus.

### 3.2. Research Platform

Two videoconferencing technologies were identified and integrated as part of the teaching platform: *Zoom* and *Meeting Owl Pro* device. These technologies were deployed in tandem and supported by different classroom setups.

The *Meeting Owl Pro* device, made by Owl Labs, provides enhanced features in videoconferencing between different parties. The device improves the effectiveness and efficiency of meetings by remotely detecting the person who is speaking as if the parties are in the same physical space. Instead of looking at computer cameras, people are free to look around and chat normally. In addition, it has the benefits of capturing subtle expression shifts and body chemistry that is often lost in a videoconferencing session when people are physically disconnected from each other [46].

The *Meeting Owl Pro* device has a conic shape and is equipped with a camera, a microphone (mic) system, and a speaker system (Figure 2). The speaker system is embedded around the device which allows the sound to be captured from all directions. The mic system (also embedded around the device) is composed of eight built-in omnidirectional mics which provide high-quality audio coverage for the entire classroom. These features allow the industry practitioners to speak/hear from students sitting at the distinct locations of the classroom. The camera is located on the tip of the *Meeting Owl Pro*, and it has a rotation of 360 degrees which allows the industry practitioners to have a panoramic view of the classroom. The camera has also the ability to focus on the student who is speaking as the mic system detects the location of the sound source, and switches focus as another student speaks. The device also has a power adapter and a USB cable that allows the connection to any computer without further configuration. It is compatible to *Zoom* as well as other videoconference platforms like *Skype* or *Google Meet*. In the *Zoom* platform, the industry practitioners have access to different display feeds on their screens. The feed split on the top of the screen shows the panoramic view of the classroom. The feed split below shows the student who is currently speaking and the student who spoke last [47].

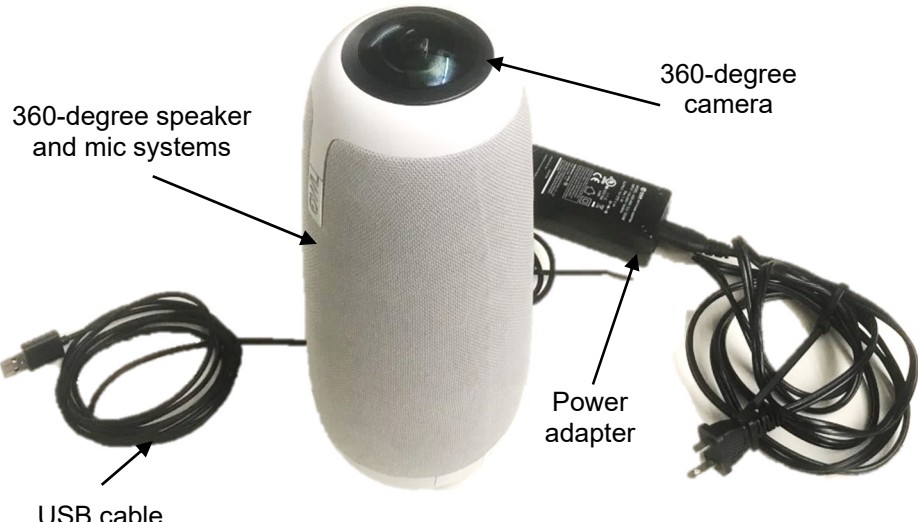

**Figure 2.** *Meeting Owl Pro* device and its different components.

The different utilities of the *Meeting Owl Pro* allowed the instructor of record to orient the classroom in different setups so that students can interact more efficiently with the industry practitioners via *Zoom*. Three classroom setups adopted to enhance the TWI model are described below.

### 3.2.1. Classroom Setup 1

Classroom Setup 1 is the standard format of the classroom where theoretical lectures were delivered. In this format, the instructor of record and the projection of the industry practitioner were positioned in front of the class, and all students were seated facing them. Since the *Zoom* projection of the industry practitioner is shown at the front screen, students can easily visualize the content (e.g., PowerPoint presentations) as well as other provided deliverables. Also, the setup facilitates the interaction among the different parties especially when students would like to ask questions to either of the instructors, or when the instructor of record would need to clarify terminologies used by the industry practitioners that are less familiar to the students. Figure 3 illustrates an actual example of this classroom setup and the schematic of this classroom organization. Objects 1, 2, and 3 are identified on the figure and detailed below.

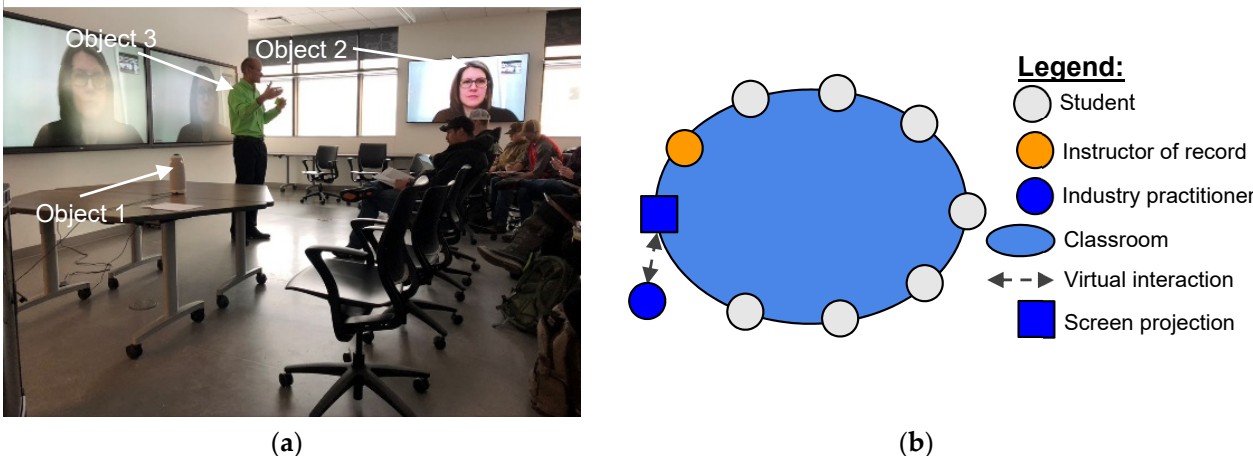

(**a**)                                                                                   (**b**)

**Figure 3.** Classroom Setup 1: (**a**) Actual example; (**b**) Classroom schematic.

Object 1: The *Meeting OWL Pro* device is placed in front of the classroom to capture the whole class.

Object 2: The industry practitioner videoconferences from her work location (office or jobsite).

Object 3: The instructor of record facilitates the teaching experience by introducing the chapter.

### 3.2.2. Classroom Setup 2

Classroom Setup 2 is the format where students were distributed in different tables throughout the classroom to have group activities in class. Within each group, open-ended questions pertaining to topics delivered by the industry practitioner are discussed. With the 360-degree feature of the *Meeting Owl Pro* device, the industry practitioner can easily participate in the discussion of each group and provide specific comments and feedback. Additional monitors were placed throughout the classroom to facilitate the interaction between the students and the industry practitioners. At the same time, the instructor of record can also roam around the classroom to provide additional instructions. Unlike in the standard format presented in the Classroom Setup 1, this setup prevents repetition of instructions for groups or students who already understood a particular problem. Figure 4 illustrates an actual example of this classroom and the schematic of this classroom organization. Object 1 is identified on the figure and detailed below.

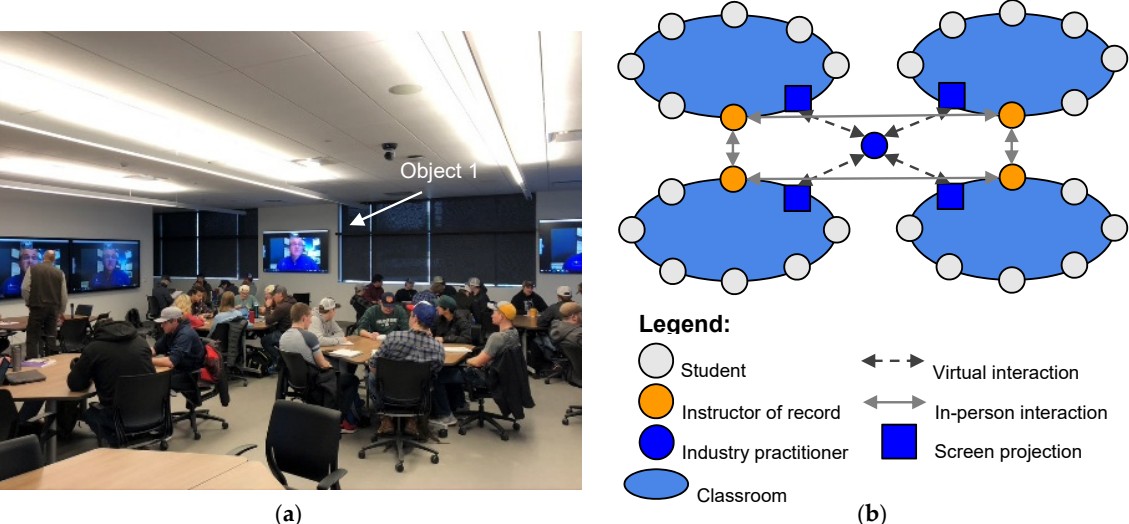

**Figure 4.** Classroom Setup 2: (**a**) Actual example; (**b**) Classroom schematic.

Object 1: The videoconferencing display of the industry practitioner is shown on all monitors throughout the classroom, which facilitates the interaction with the students.

### 3.2.3. Classroom Setup 3

Classroom Setup 3 is the virtual classroom setup (digital space) where all the parties (students, industry practitioner, and instructor of record) involved in the TWI model are virtually connected. Due to the COVID-19 pandemic and consequent restrictions on in-person meetings, the virtual classroom format was adopted to actuate the teaching of the class. In this format, students can visualize the presentations and participate in group discussions remotely. The group discussions can still be conducted efficiently since *Zoom* has a breakout room feature where students of different groups are assigned in different sessions. This feature also allowed the industry practitioner and the instructor of record to join individual sessions so that specific instructions and feedback can be given to each group. In addition, the virtual classroom setup allowed the industry practitioner to invite other industry representatives to participate and provide more insights about the industry to the class. Figure 5 illustrates an actual example of this classroom setup and the schematic of this classroom organization. Objects 1, 2, 3, and 4 are identified on the figure and detailed below.

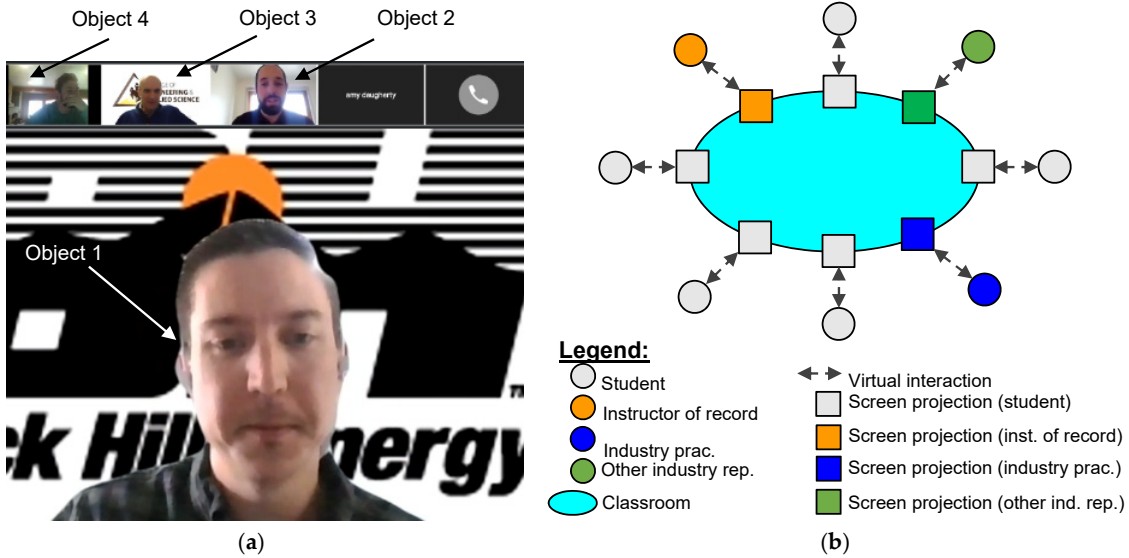

**Figure 5.** Classroom Setup 3: (**a**) Actual example; (**b**) Classroom schematic.

Object 1: The industry practitioner joins the virtual classroom to steer the class.

Object 2: The industry practitioner invites an industry representative to provide more insights about the industry to the class.

Object 3: The instructor of record joins the virtual classroom to provide further assistance to the students.

Object 4: A student joins the virtual classroom to attend the class and participates in class discussions.

### 3.3. Direct Assessment

The evaluation of the students' performance in class was conducted by way of quizzes and reflection papers, which were developed by the instructor of record, in addition to some content provided by the industry practitioners. Quizzes consisted of multiple-choice and true or false questions. Reflection papers were assigned after the industry practitioner presentation and consisted of a two-page write-up on the covered content. These reflections follow the 4MAT Learning Cycle Model developed by McCarthy [48]. The Learning Cycle Model was developed to engage students in a more dynamic manner of learning and remembering, and four questions need to be answered:

(1)    Why? Valuing new learning that connects to the learner.
(2)    What? Structuring knowledge into coherent ideas and concepts.
(3)    How? Approaching previously gained knowledge in a new or different way.
(4)    What if? Applying the gained knowledge to new problems.

In the assigned reflection paper, the first three questions of the 4MAT Learning Cycle Model were included, and students were required to do the following: produce a one-page executive summary on the material covered by the industry practitioner, which corresponds to the second question (What?) of the Learning Cycle Model; write a half-page opinion/feedback about the discussed topic and potential connections with their personal academic or work experiences, which corresponds to the first question (Why?) of the Learning Cycle Model; and come up with three questions related to the covered content, which corresponds to the third question (How?) of the Learning Cycle Model. It was decided not to include the fourth question (What if?) of the Learning Cycle Model because it was difficult to evaluate in a written assignment.

### 3.4. Indirect Assessment

To evaluate the effectiveness of the TWI model supported by using new videoconferencing technologies and innovative classroom setups, end-course surveys were administered to the students and industry practitioners. A seven-question online survey was administered to assess students' perceptions on the effectiveness of this model. Answers for questions 1 to 6 required students to select an option in a Likert scale from 1 to 5, with 1 being "*Very Unsatisfactory*", 2 being "*Unsatisfactory*", 3 being "*Neither Satisfactory nor Unsatisfactory*", 4 being "*Satisfactory*", and 5 being "*Very Satisfactory*." Question 7 was open-ended which allowed the students to express their suggestions for improvements or expressing their concerns about certain facets of the model. To assess the perception of the industry practitioners on this model, an eight-question survey was administered to all participants with specific reference to their involvement in class preparation (time commitment), class delivery (preferred teaching modality), interaction with students, future collaboration, etc.

## 4. Results and Findings

### 4.1. Direct Assessment Results

A total of eight out of fifteen chapters were evaluated through quizzes in which two of them were delivered through the TWI model supported by using new videoconferencing technologies and innovative classroom setups (Chapter 3–Accident Causation Theories, and Chapter 8–Hazardous Waste).

Figure 6 shows the average score distribution (out of 100) of the class for each chapter. The average score for all chapters was 82. It is possible to see in Figure 6 that students received relatively lower grades in the chapters taught using the TWI model (both below the average of all chapters) in comparison to the chapters taught in a conventional class. In fact, Chapters 3 and 8 were the chapters in which students obtained the second lowest and the lowest (tied with Chapter 7) scores, respectively. Although this may seem counterintuitive, the results are actually expected because there is a disconnect between the proposed type of evaluation (developed by the instructor of record) and the teaching content delivered by the industry practitioner.

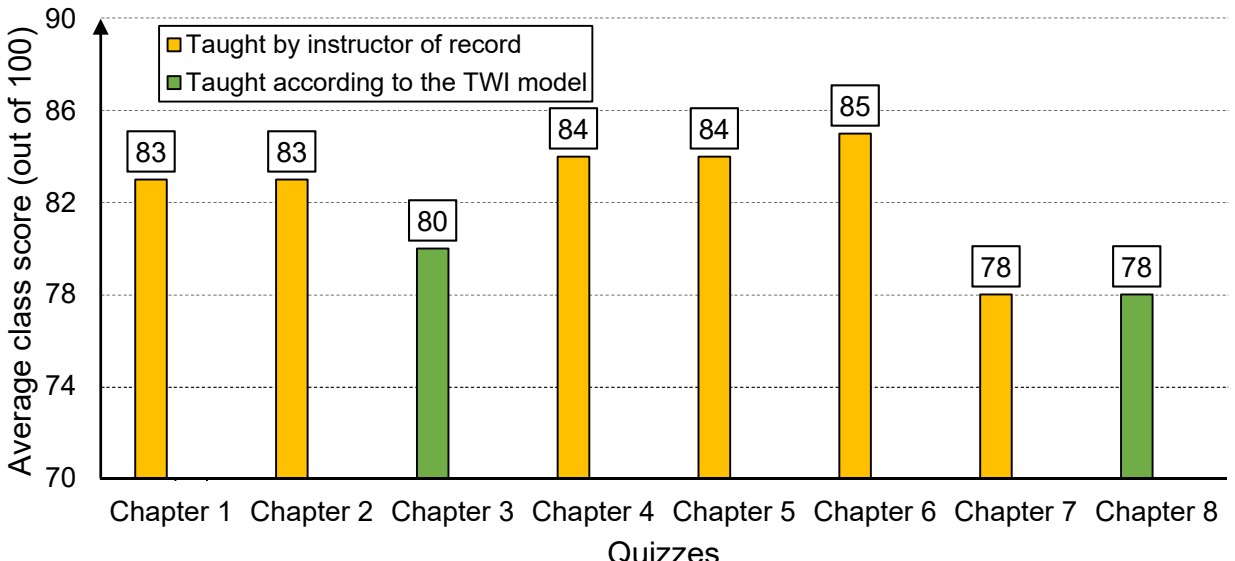

**Figure 6.** Average class score distribution for quizzes.

Because the quizzes were developed by the instructor of record, questions focused on the theoretical aspects of the content taught by the industry practitioners. Questions on the lower levels of understanding in the Structure of the Observed Learning Outcome (SOLO) Taxonomy described in Biggs and Collis [49] are typically asked (Unistructural or Multistructural). Select the correct answer for a given statement, select the word that makes a given sentence true, or identify the option that does not apply in a certain situation are some of the examples. On the contrary, the teaching methods adopted by the industry practitioners in the TWI model pertain to the practical application of the concepts. During the class discussions, real-life problems were proposed, and students were expected to come with potential solutions regarding to different conditions in the jobsite. Questions such as "*come up with a fire safety prevention outline*," "*list three unsafe acts in the field adopted by workers without proper training*," or "*how the planning of a project can affect jobsite safety*" were some of the problems proposed to the students. To answer these questions, students are required to critically think about the problems based on the delivered content and propose possible answers that suit a given scenario. This type of exercise is completely distinct from single answer questions described above, as it stimulates higher levels of understanding in the SOLO Taxonomy (Relational or Extended Abstract). Because of this difference, the quiz scores were lower for the chapters taught using the TWI model. Conversely, Figure 6 shows that the scores obtained in chapters only taught by the instructor of record were higher. This observation is expected because the type of evaluation (multiple-choice and true or false questions) matches with the way the class was taught, i.e., delivery of conceptual knowledge based on textbooks. Although some students were happy with their scores in these classes, others would have preferred to learn more about the practicality of the content rather than solely obtaining academic achievement.

For comparison purposes, the quiz grades of Chapters 3 and 8 were obtained for Spring 2020 in which industry practitioners were not integrated (quizzes administered in Spring 2020 and 2021 were almost identical). The class average was 89 and 91, respectively, which confirms the disconnect between the type of evaluation and teaching methods observed in the TWI model.

The scores obtained on the reflection paper were much higher (the task was only assigned for Chapter 3). The class average was 92 (out of 100) with only three students missing the assignment. On the one-page executive summary (What? of the Learning Cycle Model), students showed a general understanding of the class as they were able to structure the summary sequentially with most of the covered topics. Since the chapter is about accident causation theories, the discussed topics include types of accidents, common causes of accident nationwide, statistics about the number of accidents and most dangerous construction subsectors in Wyoming, and prevention strategies. Because most students identified these topics in their summaries, the scores in this section were-in general-uniform, and factors used to distinguish a full-score answer were based on grammar, fluidity of writing, and formatting.

On the half-page opinion/feedback portion of the paper (Why? of the Learning Cycle Model), students expressed their enjoyment in attending the presentation from the industry practitioner. Many liked the fact that the topic was explained from an industry point of view that students are not yet familiar with. Regarding to the information presented in the class, the students were surprised with the statistics shown in the presentation. Many were not aware of that Wyoming is one of the states that has the highest incident rate in the construction sector. The information presented in this talk allowed the transitioning of the students' thinking from a focus mode (contemplation of problems according to familiar thought patterns) to a diffused mode (establishing bigger-picture connections based on existing knowledge) [50], as they realized the importance of implementing safety protocols in the workplace.

Despite the general good performance in this assignment, many points were deducted when the students were asked to relate the presented content with their personal experience. Several students neglected this component in their papers, and others only provided abstract ideas without integrating them with their own experiences. The inability of correlating the topic with their personal experiences is expected to be a consequence of the conventional teaching methods implemented in the classes, where the focus is only on the concepts without much critical thinking. Nevertheless, a few students were able to relate the safety concerns with situations that occurred in their internships and summer jobs that they had not been aware of previously.

On the three questions related to the covered content (How? of the Learning Cycle Model), students expressed their awareness of engagement with the topic of workforce safety. Some interesting questions produced by the students are shown below:

- "Are there any penalties/incentives stimulus for companies that do not comply/comply with safety regulations?"
- "Why does Wyoming have the highest incident rate even though it has the least amount of population?"
- "What are the post-injury procedures that most organizations instill in their workers?"
- "When cultivating a safe workplace, what are the attributes and prerequisites that employers should be looking for?"
- "Is there any form that workers can fill out to report any potential aspect that does not conform with the safety standards in their workplace?"

Although the instructor of record may have been lenient in the grading of these reflection papers, as the answers are relatively subjective, students had a good overall performance in this assignment. Regardless the scores, the students were able to develop skills otherwise seldom evaluated in academia, such as critical thinking and self-reflection, which are practice-oriented competences essential to solve real-life problems in the industry.

### 4.2. Indirect Assessment Results

#### 4.2.1. End-Course Surveys Administered to Students

A total of 57 (out of 60) students successfully completed the online survey administered at the end of the semester which equates to a participation rate of 97%. The following is the breakdown of student response patterns to some of the questions asked in the survey, along with supportive comments obtained from class. Although other questions were not analyzed in this study (as the collected information is beyond the scope of this research), the list of questions asked in the survey and a graphical representation (for questions 1 to 6) with the students' positive feedback percentages (Figure A1) are included in the Appendix A (in all questions, it is considered positive feedback when students rate the statement with either a 4–"*Satisfactory*" or 5–"*Very Satisfactory*").

On the first question, students were asked to rate their impression of using *Zoom* and *Meeting Owl Pro* technologies to support presentations made by industry practitioners in the TWI model. The survey results illustrated in Figure 7 show that 77% of the students (44 students) had positive feedback on using these technologies. Clarity of sound and video imagery as well as ability to focus on different participants at the same time, are features that the students commended on the use of *Meeting Owl Pro*. In addition, students highly valued the breakout room sessions enabled by *Zoom*. 19% of the students (11 students) were neutral about using *Zoom* and *Meeting Owl Pro* to host presentations because their personal preference is to attend these classes in person. 4% of the students (2 students) did not have a good experience of using *Zoom* and *Meeting Owl Pro* because they were not familiar with these technologies. Especially during the classes delivered using Classroom Setup 3 (virtual classroom), it was observed that a few students struggled with the different functionalities such as asking permission to talk and entering a breakout room session. However, these challenges only occurred with a small percentage of the population as nowadays, university students are generally well-versed with the use of these technologies.

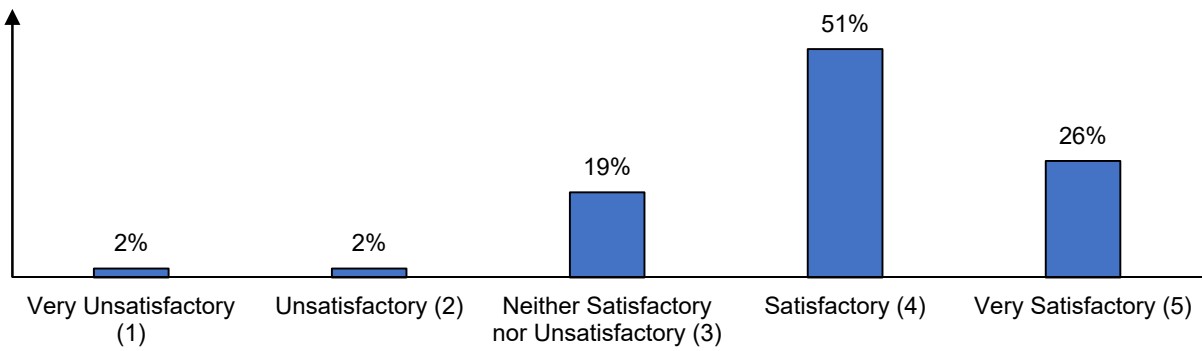

**Figure 7.** Results of the students' impression on using Zoom and Meeting Owl Pro to support the TWI model (percentages rounded to the whole unit).

On the second question, students were asked to rate their impression of using different classroom setups to support the TWI model. The survey results illustrated in Figure 8 showed that 80% of the students (46 students) had positive feedback. Students favored the use of different classroom setups as each setup (Classroom Setup 1 for lecture, Classroom Setup 2 for in-class group discussion, and Classroom Setup 3 for virtual classroom) was organized to be more efficient to a particular classroom activity. Also, students appreciate the fact that, during the group discussions, industry practitioners were able to zoom in on questions and suggestions specific to each group thanks to the 360-degree rotation of the *Meeting Owl Pro* device and its enhanced camera, mic system, and speaker system. The remaining 20% of students (11 students) were either neutral or unfavorable to the use of these innovative methods. Like the previous question, preference to in-person meetings and unfamiliarity with *Zoom* and *Meeting Owl Pro* device are possible reasons for this negative feedback.

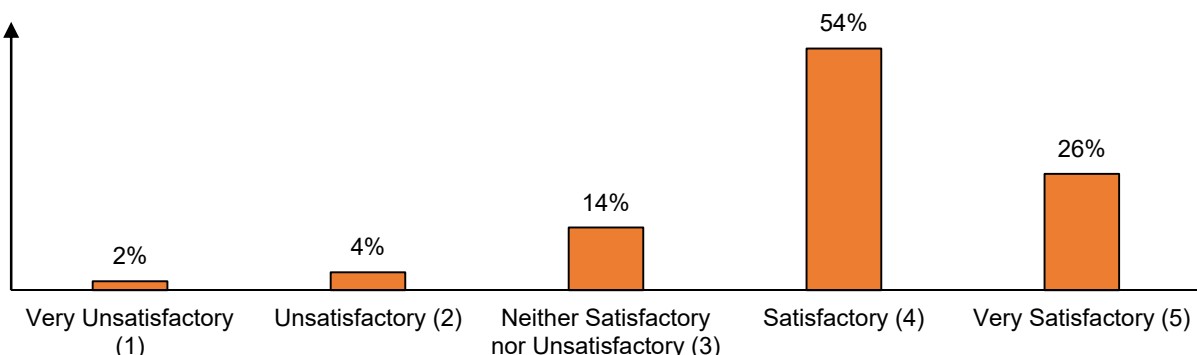

**Figure 8.** Results of the students' impression of using different classroom setups to support the TWI model (percentages rounded to the whole unit).

On the fourth question, students were asked to rate their impression of the relevance of the content taught by the industry practitioners. The survey results illustrated in Figure 9 showed 82% of the students (47 students) had positive feedback on the delivered content. Students told the instructor of record that they highly value real-life examples given by the industry practitioners as they were able to understand the practical usefulness of the knowledge acquired in the university. Other insights related to the most up-to-date practices and technologies, which are generally not included in the course materials nor textbooks, were also elaborated in these classes. In comparison to the classes taught conventionally, students pronounced that they would usually pay more attention to the industry practitioners and their industry representatives. Additionally, it was also observed that students tend to be more punctual when industry practitioners are involved. Note that in Figure 9, none of the students rated this question as 1–"*Very Unsatisfactory*" which shows the value of the content delivered in the TWI model. However, 18% of the students (10 students) were either neutral or unfavorable to the content delivered by the industry practitioners. A possible explanation for this observation is that this group of students might not be interested in the topics covered in class since they show preference to work on other areas in the construction sector.

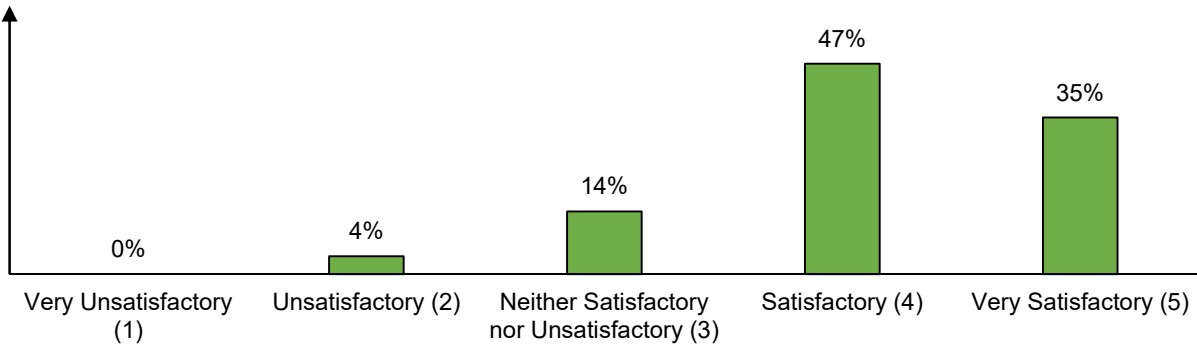

**Figure 9.** Results of the students' impression about the relevance of the content taught by the industry practitioners (percentages rounded to the whole unit).

On the seventh (last) question, students were asked to provide any comments or feedbacks on the class and especially about the TWI model supported by using new videoconferencing technologies and innovative classroom setups. In general, responses to this question were very positive. Some example responses are shown below:

— "Industry practitioners contributed greatly to the teaching platform. Would love to see that continue."
— "I am very appreciative that we were able to have industry practitioners come to class and help connect what we were learning in the classroom to how it applies in the real world."

- "The presentations were good. They brought real world examples into the classroom and gave us a feel for industry."
- "I really enjoyed having industry practitioners come in. It really shows how learning from the books interpret real life situations."
- "I thought it provided us a unique perspective that we couldn't get from the textbook."

4.2.2. End-course Surveys Administered to Industry Practitioners

All six industry practitioners completed the online survey administered at the end of the semester. A succinct analysis of their responses in addition to supportive comments obtained from meetings out of the classroom is given below. The list of questions asked in the survey is included in the Appendix A.

All the industry practitioners commented that they had a positive experience, with most of them hoping to be able to teach their classes in-person at some point in the future. Their observations were that students asked good questions, but they felt they could have had better interactions face-to-face. Although all the practitioners were comfortable with the *Zoom* and *Meeting Owl Pro* technologies used in class to enhance the TWI model, many felt that the questions could have been better answered, especially with the hands-on aspects of the coursework. However, they viewed these technologies as valuable resources to assist the teaching of a class during the COVID-19 pandemic and when the practitioners are not able to be physically present in a classroom.

All industry practitioners said they are willing to continue being involved with the TWI model, as it gave them a chance to speak about their daily tasks to the students, who are generally unfamiliar with the practices in the industry. Some of the practitioners would like to be involved in more classes while others are happy to teach the same course the next time it is offered.

The time that took the industry practitioners to prepare their respective presentations varied. For some, it took more than 3 h and for others, it took less than 1 h. This variation depended on the experience these practitioners have with university students. For many, this is not the first time as they have experience presenting seminars at other universities. Others, on the other hand, needed more time to prepare their presentations and practice their speech so that the content is more suitable for the students' comprehension. Some suggested that additional guidance could be provided by the administrators of the program, especially for first-time industry practitioners involved in the TWI model.

**5. Conclusions and Future Recommendations**

Based on theoretical evidence, many applied-based degree programs face difficulties in bridging the gap between traditional classroom instruction and skills and competencies required in the industry. A popular solution that has been adopted by many institutions is to involve industry practitioners for the co-teaching of courses in the curriculum. This model is called "Teaching with Industry" (TWI). To enhance the TWI model for when the industry practitioners cannot be physically present, additional features can be added to increase the students' learning, such as using new videoconferencing technologies like *Zoom* and *Meeting Owl Pro*, and innovative classroom setups. This enhanced TWI model was put into practice in a Construction Management course (CM 2300: Construction Safety) taught in Spring 2021 at the University of Wyoming.

In this class, two out of fifteen chapters were taught using this TWI model, with a total of six industry practitioners involved. On those chapters, the industry practitioners were responsible for developing their own presentations and providing insightful information about their work experience, industry skills, and real-life examples. *Zoom* and *Meeting Owl Pro* were used as technologies to enhance the interaction between students and industry practitioners during class. The *Meeting Owl Pro* device is equipped with a camera, a mic system, and a speaker system that capture images and sound in 360 degrees. The camera has the ability to focus on a student who is speaking as the mic system detects the location of the sound source, and switches focus as another student speaks.

The integration of different classroom setups in the TWI model helped the students to interact more efficiently with the industry practitioners. Three different classroom setups were adopted for different activities carried out in class. The setups were adopted for the delivering of presentations, group activities and discussions, and receiving instructions and feedback specific to each group.

Direct assessment showed that students actually obtained lower scores on quizzes when the class was taught using the TWI model in comparison to classes taught in a conventional fashion. This observation was expected due to the disconnect between the teaching and evaluation methods: the quizzes only focused on conceptual aspects of the class content (multiple-choice and true or false questions developed by the instructor of record), while the industry practitioners emphasized the practical application of concepts in real-life problems. Conversely, students performed well on the reflection paper about Chapter 3–Accident Causation Theories taught using the TWI model. This observation was also expected as the teaching aligns with the evaluation which required the students to think critically about the covered content and come up with their own interpretation. Although students did not have difficulties in producing the executive summary and questions related to the covered content, many felt challenged in trying to correlate class content with their personal experiences.

End-course surveys administered to the students showed that most of them gave positive feedback on using *Zoom* and *Meeting Owl Pro*. Although many highly praised the features offered by these technologies such as the sound and video clarity, and the ability to divide the class in breakroom sessions using *Zoom*, a few students did not enjoy their experience as they struggled with the technologies and preferred to attend the class in person. Most of the students also commended the use of different classroom setups to support the TWI model, as each setup was organized to be more efficient to a specific activity. In particular, the student appreciated the individual feedback received during group discussions from the industry practitioners thanks to the 360-degree rotation of the *Meeting Owl Pro device*. The majority of the students also expressed their positive feedback on the relevance of the content taught in the TWI model. The students highly valued the real-life examples taught by the industry practitioners as they get to understand the practical usefulness of the knowledge acquired in the courses taught at the university.

End-course surveys administered to the industry practitioners showed that all of them had a good experience of being part of the TWI model. Although they viewed the use of *Zoom* and *Meeting Owl Pro* technologies as valuable resources to assist the teaching of a class during the COVID-19 pandemic, they would prefer a face-to-face interaction, especially to explain hands-on aspects of the coursework. In addition, all industry practitioners expressed their willingness to continue being involved with the TWI model. The amount of time that took industry practitioners to prepare for the class varied between less than 1 h and more than 3 h. Industry practitioners that spent less time already had experience with university teaching, while the others needed time to prepare their presentation and speech to better suit the students' understanding.

The enhanced TWI model described in this paper takes advantage of the combined benefits of the three components: participation of industry practitioners in the curriculum teaching, use of new videoconference technology, and employment of different classroom setups, in order to provide a better teaching/learning experience to all the entities. With this combination, in-class activities that stimulate students' underdeveloped soft skills can be conducted efficiently. Moreover, insights regarding to practical examples can still be acquired efficiently even when industry practitioners are not physically present in the classroom. The outcomes of the direct and indirect assessments in this case study confirm the students' lack of industry exposure frequently reported in the literature. Nevertheless, most of the students showed willingness to be involved and learn more about the practice of the profession. Therefore, it is up to the program administrators to develop TWI models like this to facilitate students' contact with industry practitioners.

Based on the observations of this case study, an improvement that could be made in the direct assessment is the replacement of single answer problems to open-ended questions. The latter would be more in alignment with the teaching methods adopted by industry practitioners (and actual problems encountered in the industry) which require students to think critically about real-life problems. Another improvement on the TWI model is to provide a standardized form that facilitates first-time industry practitioners preparing their presentation. This form would contain guidelines and recommendations given by more experienced industry practitioners of the TWI model so that the presentations can be developed to better suit students' comprehension.

The authors recognize the limitations of this study as (1) it only evaluated this enhanced teaching model according to data obtained for a single class, and (2) the indirect assessment results were not validated by directly comparing to other classes in which the model was not implemented. Nevertheless, due to the success observed in the CM 2300: Construction Safety course of Spring 2021, the Construction Management program of the University of Wyoming is planning to expand the TWI model to at least one-third of the courses in the curriculum in order to bridge the gap between academia and industry by the end of 2022. Additional direct and indirect assessment data will then be collected so that more improvements to the teaching model can be made. Furthermore, it is expected that later developments of the TWI model will yield more appropriate measures and methods of assessment. Adopting the current model as a baseline, the new collected results will be used to further improve this teaching model so that students are better prepared for their lives in the practice of the profession.

**Author Contributions:** Conceptualization, F.J. and W.C.; methodology, F.J., W.C., R.L. and A.D.; formal analysis, F.J., W.C., R.L. and A.D.; resources, F.J., W.C. and R.L.; data curation, R.L. and A.D.; writing—original draft preparation, R.L.; writing—review and editing, F.J., R.L.; supervision, F.J., W.C., R.L. All authors have read and agreed to the published version of the manuscript.

**Funding:** This research received no external funding.

**Institutional Review Board Statement:** The study was conducted in accordance with the Declaration of Helsinki, and approved by the Institutional Review Board (or Ethics Committee) of the University of Wyoming (protocol code 20200330FJ02721; date of approval: 16 April 2020).

**Informed Consent Statement:** Informed consent was obtained from all subjects involved in the study.

**Data Availability Statement:** The data used in this study is available from the corresponding author by reasonable request.

**Conflicts of Interest:** The authors declare no conflict of interest.

## Appendix A

List of questions on the end-course survey administered to students:

(1) Rate your overall impression of using *Zoom* and *Meeting Owl Pro* to support the presentations delivered by industry practitioners.
(2) Rate your overall impression of the different classroom setups to support the class activities involving industry practitioners.
(3) Rate the overall quality of the content of the presentations delivered by industry practitioners.
(4) Rate the overall relevance of the content of the presentations delivered by industry practitioner (e.g., *"Did the presenters offer valuable insights from the field?"*).
(5) Rate how well the presentations delivered by the industry practitioners were organized (e.g., *"Did it start on time?"*, *"Did the presenter seem confident in presenting* via *Zoom?"*).
(6) Rate your impression of the value of the industry insights offered in the presentations delivered by industry practitioners.

(7) Please write down other impressions and observations you had on the industry practitioners.

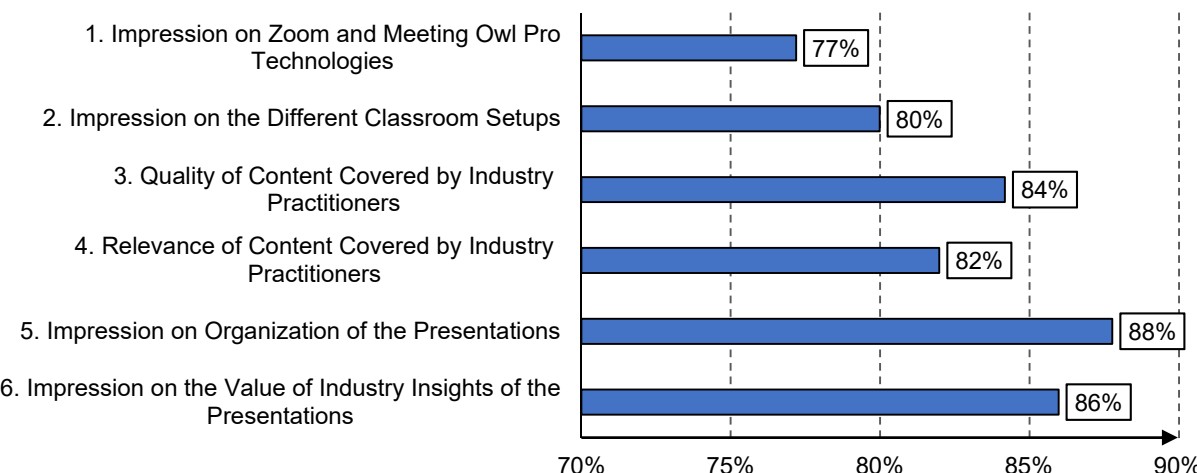

**Figure A1.** Percentages of the positive feedbacks collected from the end-course survey administered to students for questions 1 to 6 (percentages rounded to the whole unit).

List of questions on the end-course survey administered to industry practitioners:

(1) How long did it take you to prepare for your class presentation?
(2) Please provide suggestions in how to prepare for future classes under the TWI model.
(3) How comfortable were you during this teaching experience? Please provide a rating between 1 (not comfortable) to 10 (very comfortable).
(4) Do you think one class (50 min) was enough to finish your assigned content? Please provide a rating between 1 (not enough) to 10 (more than enough).
(5) Please provide suggestions about class delivery (e.g., PowerPoint templates, guidelines, etc.) that might be helpful for the preparation of future presentations.
(6) Please provide suggestions in how to improve interaction with students under the TWI model (*e.g., prior reading, group work, etc.*).
(7) Would you like to continue your TWI engagement? If yes, how many classes would you like to teach in the future?
(8) How do you rate your teaching experience? Please provide a rating between 1 (not good) to 10 (excellent).

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
