# Peer review of "Case Study: Teaching with Industry (TWI) Using New Videoconferencing Technology and Innovative Classroom Setups"

_education, doi:10.3390/educsci12020128_

Round 1
Reviewer 1 Report
Dear authors. It seems to me an interesting proposal on how to approach the training process linked to the use of multimedia supports for the retransmission of classes through video platforms.However, a more empirical approach to publication seems to me absolutely necessary. It would be necessary to compare the academic performance of students who use these media and others who do not use them to see the real impact of the use of these platforms (control and experimental group).
It would be necessary for this assessment to be made with a statistically validated instrument, with exploratory and confirmatory factor analysis for its application.
If we cannot make that comparison of the improvements or non-improvements with the use of these means between students who use it and others who do not, as well as if the instruments are not validated for analysis, the study responds only to the explanation of an experience not empirically generalizable for application by other researchers.
I think that reformulating the original idea, it is a proposal that can provide interesting data, but it needs that comparison and a solid and statistically validated instrument.
Best regards
Author Response
Response: The authors appreciate the feedback. However, because the program is newly founded, no statical data was collected in the first year when the CM 2300: Construction Safety class was taught. On the following years, the participation of industry practitioners was then incorporated in some of the chapters. For comparison, the authors first evaluated the scores obtained in the quizzes (direct assessment) with and without industry practitioners. Although it may seem counterintuitive, as the results in the classes taught by industry practitioners were lower, it makes sense because the quizzes were still developed by the instructors of record based on the textbooks with multiple-choice and true or false questions. In the updated version of the manuscript, quiz grades from the first year in which there had been no participation of industry practitioners was added. The results confirm this explanation.
On the indirect assessment surveys, although the results are not directly compared to a control group, every innovative element of the model: participation of industry practitioners, videoconferencing technology, and different classroom setups, were discussed based on explanations for students who liked and did not like it. More details were added to emphasize this comparison in the updated version of the manuscript.
In the authors’ opinion, the comparison of a class that does not involve participation of industry practitioners and does not use videoconferencing technology nor different classroom setups, is a conventional class taught in any academic institutions. The problems observed in these are already commonplace in the existing literature. These problems were also the reason that prompted the development of this innovative teaching model. Nonetheless, the authors added in the updated version of the manuscript that data from a control group (in which this teaching model is not implemented) could be collected in future studies as an expansion of the current research.
Reviewer 2 Report
The text is very interesting and addresses a very innovative idea and the study carried out is very well written and explained. It is only considered that the description of the results and/or the conclusions could have been more articulated with the theoretical body of the work.
It is also considered that the references could be a little more current since more than 70% of them are more than 5 years old.
Author Response
Response: The authors appreciate the feedback. Please see the updated manuscript.
More explanation was added throughout the updated version of the manuscript (more concentrated on the “Direct Assessment Results” and “Indirect Assessment Results” sections) to connect the results and conclusions to the theorical body of work.
Additionally, some of the references were added/replaced to others more recent in the updated version of the manuscript.
Reviewer 3 Report
- The research gap is clear. However, the problem's background is unclear. Thus, I suggest that the authors explain more about the problems background and its statement.
- The authors should add the theoretical and practical implications of this research at the end of the manuscript.
- The authors should show the limitations of this research.
Author Response
- The research gap is clear. However, the problem's background is unclear. Thus, I suggest that the authors explain more about the problems background and its statement.
- The authors should add the theoretical and practical implications of this research at the end of the manuscript.
- The authors should show the limitations of this research.
Response: The authors appreciate the careful review of the paper. Please see the updated manuscript and below for a detailed response to each comment.
- More details were added in the updated version of the manuscript to clarify problems caused by the disconnect between Academia and Industry. This was added in the “Introduction” section as the following:
“The gap between classroom instruction and industry requirements can be especially noticeable in career and technical education (CTE) degrees, like Construction Management. Much of what is expected and required in students of CTE degree programs may not translate easily into desired skill sets for employers. Industry recruiters observed that although new hires generally have sufficient technical skills, many do not have a good grasp of the actual applicability of these skills in an actual project [4]. Others seem to have lack of soft skills such as critical-thinking, leadership, and communication, which are as important as the technical competencies [5]. Some employers believe that the practices taught in academia are obsolete and too didactic which have little practical use in the practice of the profession [6]. Thus, a small group of employers are even willing to hire experienced personnel without a four-year college degree in lieu of freshly graduated students without experience [7]. As a matter of fact, many students do not feel prepared to transition to the professional market. A great number think that the theoretical component learned in higher education is not enough to gain advantage in the construction industry [8]. Also, students with industry work experience reported that they did not have the capabilities at the beginning to carry out some of the proposed tasks on the jobsite [9]. Many needed additional training before being assigned to actual tasks from their representing companies [10].”
- The theoretical and practical implications of this research were added to the “Conclusions and Future Recommendations” section of the updated version of the manuscript as the following:
“The enhanced TWI model described in this paper takes advantage of the combined benefits of the three components: participation of industry practitioners in the curriculum teaching, use of new videoconference technology, and employment of different classroom setups, in order to provide a better teaching/learning experience to all the entities. With this combination, in-class activities that stimulate students’ underdeveloped soft skills can be conducted efficiently. Moreover, insights regarding to practical examples can still be acquired efficiently even when industry practitioners are not physically present in the classroom. The outcomes of the direct and indirect assessments in this case study confirm the students’ lack of industry exposure frequently reported in the literature. Nevertheless, most of the students showed willingness to be involved and learn more about the practice of the profession. Therefore, it is up to the program administrators to develop TWI models like this to facilitate students’ contact with industry practitioners.”
- Limitations of this research was added to the “Conclusions and Future Recommendations” section of the updated version of the manuscript as the following:
“The authors recognized the limitations of this study as (1) it only evaluated the validity of this enhanced teaching model for a single class, and (2) the results were not directly compared to other classes in which the model was not implemented. Nevertheless, due to the success observed in the CM 2300: Construction Safety course of Spring 2021, the Construction Management program of the University of Wyoming is planning to expand the TWI model to at least one-third of the courses in the curriculum in order to bridge the gap between academia and industry by the end of 2022. Additional direct and indirect assessment data will then be collected so that more improvements to the teaching model can be made. Furthermore, a survey regarding to students’ learning will be administered to classes in which this model is not implemented. This feedback will be used as a control group to further demonstrate the importance of integrating industry participation in academia.”
Reviewer 4 Report
Overall, the paper is interesting and investigates how online teaching can be used in a timely and relevant way for both research and teaching. The discussion is well conducted and based on relevant literature. However, I find a few issues with the paper that I hope the authors can address:
- The results are presented in a less than optimal manner. The section starts out by mentioning all the stuff that has not been done and then comes a figure which tells that the grade was better for the activities not included. However, as I understand the text after a few readings, some data (reflection paper) are not included in the figure and so forth. The authors can improve this part of the story.
- Direct assessment results are the heading twice - the second is indirect?
- The role of the reflection paper is rather exciting and could maybe be explained a little further. Did the students obtain higher taxonomical levels (Solo Taxonomy), did it aid their defuse mode thinking(A mind for numbers, Barbara Oakley), or did it give them a new way of expressing their skills? This discussion could lift the paper a slight bit.
- There are some clumsy formatting errors (e.g. line 375 and 383)
Author Response
Response: The authors appreciate the careful review of the paper. Please see the updated manuscript and below for a detailed response to each comment.
- The authors admitted that the explanation was a little ambiguous. What the authors originally meant about the exam evaluation is that it was not possible to discretize the effects of this enhanced teaching model to the effects of conventional teaching. Although questions regarding to the content that industry practitioners taught were asked in these exams, it is rather difficult to classify the type of question (i.e., taught by industry practitioners or taught in a conventional class). Because of this ambiguity, the authors decided to remove the evaluation through exam from the updated version of the manuscript.
On the quiz scores, the point was to demonstrate that there is an apparent contradiction between the students’ grades and involvement of industry practitioners in the class (decrease in the students’ quiz grades). This is later found to be expected as an indication of the disconnect between the teaching method and the type of evaluation. The industry practitioners do not teach concepts but rather examples and experiences, whereas the quizzes were still developed by the instructors of record which focus mainly on the theory. This was confirmed by comparing the quiz results taught in the previous year when industry practitioners were not involved (added to the updated version of the manuscript).
On the other hand, this disconnect was not found in the reflection paper because the questions asked in this evaluation are more directed to the presentations delivered by the industry practitioners. However, students struggled on trying to connect their personal experiences with what was presented because they are not used to evaluations that require critical thinking.
- Thank you for catching the typo. The second heading was changed to “Indirect Assessment” in the updated version of the manuscript.
- The authors really appreciate the insightful sources of information.
Oakley (2014) was added to the explanation of the reflection paper in the updated version of the manuscript as the following:
“The information presented in this talk allowed the transitioning of the students’ thinking from a focus mode (contemplation of problems according to familiar thought patterns) to a diffused mode (establishing bigger-picture connections based on existing knowledge) [50], as they realized the importance of implementing safety protocols in the workplace.”
Additionally, the authors felt more comfortable in adding Biggs and Collis (1982) in the explanation for the quiz scores with the intent to differentiate the levels of understanding approached in the class taught by the industry practitioners versus the actual evaluation. This explanation was added to the updated version of the manuscript as the following:
“Because the quizzes were developed by the instructor of record, questions focused on the theoretical aspects of the content taught by the industry practitioners. Questions on the lower levels of understanding in the Structure of the Observed Learning Outcome (SOLO) Taxonomy described in Biggs and Collis [45] are typically asked (Unistructural or Multi-structural).”
“This type of exercise is completely distinct from single answer questions described above, as it stimulates higher levels of understanding in the SOLO Taxonomy (Relational or Ex-tended Abstract). Because of this difference, the quiz scores were lower for the chapters taught using the TWI model.”
- The “blanks” left on lines 375 and 383 are supposed to show the state of our affiliation. For a double-blind review process, we thought in removing that information.
Round 2
Reviewer 1 Report
Dear authors.The application of the model is very innovative, there is no doubt about that and it is worth highlighting.
However, the original problem is that if we do not confirm the real impact of an innovation in the learning process, it would not make pedagogical sense.
I appreciate the efforts to explain the results between the first years, specifying the data collected and the forecast of making the comparative analysis.
The instrument is still not validated and no improvements have been made in this regard, being one of the notorious problems of the work.
However, I give a positive assessment for innovation, although the subject of the instrument should have been addressed for validation.
Author Response
Dear Reviewer 1, we appreciate your ongoing supportive feedback, please see attachment in support of our response.
Best

Reviewer 4 Report
The authors have done a good job updating the article. Upon reading it again, I have not found any further comments, except for a clarifications.
Line 349-351, should the assigned questions be in italic, like later in the text?
Author Response
Dear Reviewer 4, we appreciate your ongoing supportive feedback, please see attachment in support of our response.
Best
